# Alien, Naturalized and Invasive Plants in China

**DOI:** 10.3390/plants10112241

**Published:** 2021-10-20

**Authors:** Aiying Zhang, Xunyu Hu, Shenhao Yao, Mingjian Yu, Zhixia Ying

**Affiliations:** 1College of Life Sciences, China Jiliang University, Hangzhou 310018, China; ayzhang@foxmail.com; 2MOE Key Laboratory of Biosystems Homeostasis & Protection, College of Life Sciences, Zhejiang University, Hangzhou 310058, China; fishmj@zju.edu.cn; 3East China Inventory and Planning Institute, National Forestry and Grassland Administration, Hangzhou 310019, China; huxunyucooky@163.com; 4Zhejiang Museum of Natural History, Hangzhou 310014, China; chemyaoshawn@gmail.com; 5School of Life Science, Nanchang University, Nanchang 330031, China

**Keywords:** America, annual, clustered, landscaping, phylogenetic

## Abstract

Invasive species are a subset of naturalized species, and naturalized species are a subset of alien species. Determining the relationship among these three assemblages would be helpful in predicting and preventing biological invasion. Here, we reviewed the families, lifeforms, origins, introduction pathways and phylogenetic diversity of alien, naturalized and invasive vascular plants in China. The results show that species in the Asteraceae, Fabaceae and Poaceae families had a high dominance among alien, naturalized and invasive species. Moreover, almost all alien species in the Amaranthaceae, Solanaceae, Convolvulaceae and Euphorbiaceae families became naturalized species, and about 26.7% of the naturalized species became invasive species. Perennial herbs comprised a higher proportion of alien species than did annual herbs, though annual herbs were more suited to becoming invasive than perennial herbs. A considerable proportion (57.8%) of invasive species were introduced from America. More than half (56.5%) of alien species were introduced for their ornamental value, and half of these have become naturalized in China. Moreover, about half (55.2%) of all invasive species were introduced for their economic value (including ornamental, foraging and medicinal purposes). Invasive species were phylogenetically clustered and phylogenetically distant from alien and naturalized species, which indicates that phylogenetic differences could be helpful in becoming invasive. There is no doubt that human activity plays a significant role in biological invasion. This study suggests that when introducing alien species to a region, decision-makers should certainly consider the species’ phylogeny, beyond just its fundamental characteristics.

## 1. Introduction

It is well-known that biological invasions have caused great economic and ecological losses all over the world [1,2,3]. Exploring the mechanisms driving species invasion has been a hot topic for decades. Understanding which kinds of alien species could easily become invasive species would help decision-makers to predict and thus prevent biological invasion.

To become an invasive species, an alien species needs to proceed along the introduction–naturalization–invasion continuum [4,5,6]. A species is considered naturalized, which is the fundamental precondition and a primary stage for invasion, when an alien species has established self-replacing populations that could persist for at least 10 years without human intervention [7,8]. On the other hand, if alien plants cannot form self-replacing populations and die out eventually in an area, they could be called as casual aliens [8]. Naturalized species could then become invasive species, but only once they reproduce reproductive offspring, often in large numbers, at considerable distances from the parent plants, and thus have the potential to spread over a large area [7]. In sum, invasive species are a subset of naturalized species, and naturalized species are a subset of alien species. Understanding the distinction among alien, naturalized and invasive species allows researchers to estimate the potential of an alien species to become invasive.

As the world’s third largest country, China has faced severe ecological disasters (such as notorious water hyacinth) and abundant economic losses caused by alien invasive vascular plants [9,10,11]. Here, we collected data on families, lifeforms, origins, introduction pathways and phylogenetic diversity of alien, naturalized and invasive vascular plants in China from various references to deepen our fundamental understanding of invasive plants, and thus to predict and prevent future biological invasion.

## 2. Materials and Methods

### 2.1. Data Collection

Our database of alien, naturalized and invasive plants is based on published references (Appendix A). The Latin names, lifeforms, origins and introduction pathways from published references were corrected according to the Flora of China (FOC, http://foc.iplant.cn/, accessed on 23 November 2019) and the Catalogue of Life China (CoLC, https://www.catalogueoflife.org/annual-checklist/2019/, accessed on 5 October 2021). After this correction, we double-corrected the Latin names following The Plant List (TPL, http://www.theplantlist.org/, accessed on 18 September 2013) for consistency of the phylogenetic tree, whose families are organized based on APG III [12]. It is worth noting that some species were identified as alien species before. However, it turned out to be native species to China. We put them in Appendix A and marked them as “Native to China” to tell the readers that why it is not an alien species. Biodiversity Information Standards (TDWG, https://www.tdwg.org/, accessed on 5 October 2021) were used to analyze the origins of alien species. Since there was only a small number of biennials, we classified biennials as perennials to simplify data analysis.

### 2.2. Phylogenetic Diversity

We built our phylogenetic tree using the package *‘V.PhyloMaker’* [12] in R 4.0.3 [13], which we also used for further analyses. The mega tree of this R package was consistent with the taxonomic system of The Plant List. The nearest taxon index (NTI), one of the commonly used phylogenetic diversity indices, is defined as the standardized effect size of the mean phylogenetic distance to the nearest taxon for each taxon (MNTD) in the assemblage [14]. NTI is calculated as: NTI = −(MNTD_observed_ − MNTD_randomized_)/(sdMNTD_randomized_), where MNTD_observed_ is the observed MNTD, and MNTD_randomized_ and sdMNTD_randomized_ are the expected mean and standard deviation of all nearest pairs of the randomized assemblages, respectively [15]. Here, the null model was conducted using function *mntd.query* in R package *‘PhyloMeasures’* [16] to maintain species richness. More specially, all the alien species were put into a reference pool. With this pool, we randomly run a fixed number of species richness for three assemblages in each 1000 simulated draws. Then, we calculated the mean and standard deviation of MNTD of the randomized assemblages. When NTI values were zero, the species were randomized assemblages. When the NTI values were positive, the observed MNTDs were less than that of the randomized assemblages, meaning that species were more closely related than would be in a randomized assemblage, i.e., species were phylogenetically clustered [14,16]. Inversely, when the NTI values were negative, the observed MNTDs were greater than that of the randomized assemblages, meaning that species were more distantly related than would be in a randomized assemblage, i.e., species were phylogenetically overdispersed [14,16].

The MNTDs between assemblages (alien, naturalized and invasive), as calculated by function *comdistnt* in R package *‘picante’* [17], were visualized using Principal Co-ordinates Analysis (PCoA) within R package *‘FD’* [18]. The first two PCoA axes for MNTD explained more than 95% of the variance. However, the first two PCoA axes for MPD (mean phylogenetic distance) showed a weakly represent (only explained less than 55% of the variance), indicating that the detected patterns might be random. Thus, we only used MNTD and NTI for all the analysis.

## 3. Results

There are 1686 alien vascular plant species (belonging to 783 genera in 137 families) in China, of which there are 1198 naturalized species (belonging to 618 genera and 122 families) of which there are 232 invasive species (belonging to 141 genera and 48 families), and 488 are casual alien species (alien but not naturalized species, belonging to 252 genera and 70 families).

Species in the Asteraceae, Fabaceae and Poaceae families showed a high dominance among alien species (Asteraceae accounting for 12.3%, Fabaceae accounting for 9.2% and Poaceae accounting for 7.1%), as well as among naturalized species (Asteraceae accounting for 13.0%, Fabaceae accounting for 11.1% and Poaceae accounting for 9.5%) and invasive species (Asteraceae accounting for 20.3%, Fabaceae accounting for 9.9% and Poaceae accounting for 12.1%, Figure 1a). Interestingly, almost all alien species in the Amaranthaceae, Solanaceae, Convolvulaceae and Euphorbiaceae families became naturalized species, and about 26.7% of these became invasive species (Figure 2). The Amaranthaceae family comprised 2.7% of alien species and 8.6% of invasive species. The Solanaceae family comprised 2.7% of alien species and 5.2% of invasive species, the Euphorbiaceae family comprised 2.1% of alien species and 4.3% of invasive species, and the Convolvulaceae family comprised 1.7% of alien species and 4.3% of invasive species. (Figure 1a). These results suggest that, in addition to the well-known Asteraceae, Fabaceae and Poaceae families, species in the Amaranthaceae, Solanaceae, Euphorbiaceae and Convolvulaceae families also tend toward becoming invasive species.

Annual and perennial herbs were prevalent among alien species (Figure 1b). The proportion of annual species increased dramatically from the alien to invasive categories, from 33.6% to 62.1%, while perennial species decreased from 49.0% to 26.3% (Figure 1b).

Most alien species were introduced from America, Europe, Africa, tropical Asia and temperate Asia, and this also held true for naturalized species (Figure 1c). However, only 3.9% of invasive species were introduced from tropical Asia (Figure 1c). It is worth noting that a considerable proportion (57.8%) of invasive species was introduced from America (Figure 1c). Among the alien species introduced from America, about 70% were introduced for their economic value, including ornamental, medicinal and foraging purposes, while only 30% were introduced by other ways, including unintentionally, naturally and unknown (Appendix A). However, among the invasive species introduced from America, about half of them were introduced by other means and the other half were introduced for their economic value (Appendix A).

More than half (56.5%) of alien species were introduced to China for their ornamental value, and half of these became naturalized (975 alien species to 518 naturalized species, Figure 1d). Only 16.7% of alien species were introduced by other means (Figure 1d). However, about half of all invasive species were introduced by other means, and the other half were introduced for ornamental, foraging and medicinal purposes (Figure 1d).

The NTI value of alien species was zero, meaning that alien species were randomly distributed (Figure 3a). The NTI values of different species assemblages were ranked as: alien < NNI (naturalized but not invasive) < casual aliens ≈ naturalized < invasive, which indicates that invasive species had the most clustered assemblages (Figure 3a). Invasive species were phylogenetically distant from other assemblages, while alien, naturalized and NNI species were phylogenetically close to each other (Figure 3b).

## 4. Discussion

It was not surprising to find that Asteraceae, Fabaceae and Poaceae comprised high proportions of alien species because they are among the biggest plant families worldwide [19,20,21]. What is more, the great economic value of these three families also facilitates their introduction and spread [22,23]. As Asteraceae contains common ornamental plants, about half of alien Asteraceae species (49.0%, Appendix A) in China were introduced for their ornamental value. About half the introduced species in the Poaceae family (42.0%, Appendix A) were brought to China as forage. Species in the Fabaceae family were mainly introduced to China for ornamental (31.0%, Appendix A) and forage (16.1%, Appendix A) purposes. Similarly, species in the Cactaceae family were all introduced for their ornamental value (Appendix A), and about half the species of the Solanaceae (40.0%) and Convolvulaceae families (57.1%) were introduced for their ornamental value as well (Appendix A). However, in the families more prone to becoming invasive, about half of the alien species in the Amaranthaceae (45.7%) and Euphorbiaceae (51.4%) families were introduced by other means (Appendix A).

Perennial herbs have been introduced more often than annual herbs, but their naturalization and invasion rates were much lower than that of annual herbs. Prentis [1] found that a species can adapt to a novel environment in 20 generations or less. Annual herbs are short-lived opportunists, growing rapidly and maturing early, and thus having a high likelihood to survive and become established in a new habitat [24]. These characteristics contribute to the advantages that annuals have in invasion.

A considerable proportion of alien, naturalized and invasive species were introduced from America, and most of them were introduced for their economic value, including ornamental, medicinal and foraging purposes. It is widely known that China has become one of the biggest trading partners of North America for decades [25], and this has promoted species introductions [9,26]. Moreover, North America has a similar climate (similar range of latitude) to mainland China, and South America has a similar climate to the Taiwan Province and the Hong Kong Special Administrative Region [27,28]. Similar climates might better facilitate the acclimation of alien species from America to their new habitats in China, allowing them to successfully colonize in a short time [29].

The present study shows that invasive species phylogenetically differ from NNI (naturalized but not invasive species), as well as casual aliens, alien and naturalized species. Even with a lack of direct evidence, this result indicates that phylogenetic differences may help a species become invasive. Similarly, Divíšek [6] has found that functional differences enhance invasion success, i.e., invasive species need to be functionally different enough from native species to become invasive.

## 5. Conclusions

The present study reviewed the families, lifeforms, origins, introduction pathways and phylogenetic diversity of alien, naturalized and invasive plants in China. Interestingly, besides the well-known Asteraceae, Fabaceae and Poaceae families, species in the Amaranthaceae, Solanaceae, Euphorbiaceae and Convolvulaceae families also tended to become invasive species in China. Moreover, this study suggests that phylogenetic differences help alien species become invasive. In other words, we should avoid introducing alien species that are very phylogenetically different from local species. However, take caution that this conclusion lacks direct evidence. Further studies should focus on assessing the phylogenetic and functional traits of alien, naturalized, invasive and local species altogether for a deeper understanding of the invasion mechanism, thus helping to protect against the danger of biological invasion.

## Figures and Tables

**Figure 1 plants-10-02241-f001:**
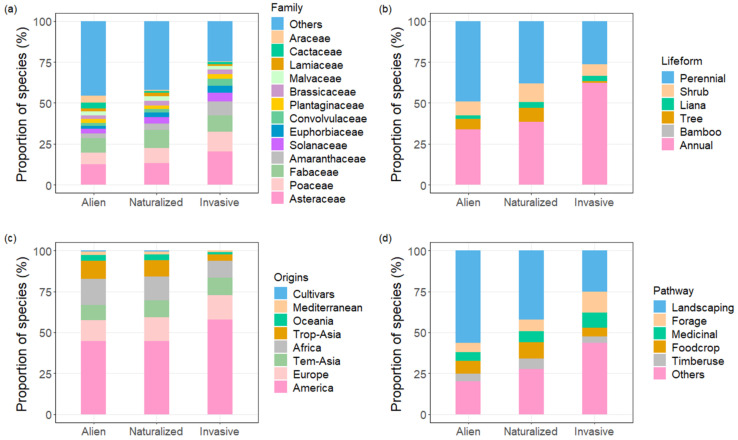
Proportions of plant species comprising alien, naturalized and invasive plants, by family (**a**), lifeform (**b**), origin (**c**) and introduction pathway (**d**). See more details for the abbreviations and categories in Appendix A.

**Figure 2 plants-10-02241-f002:**
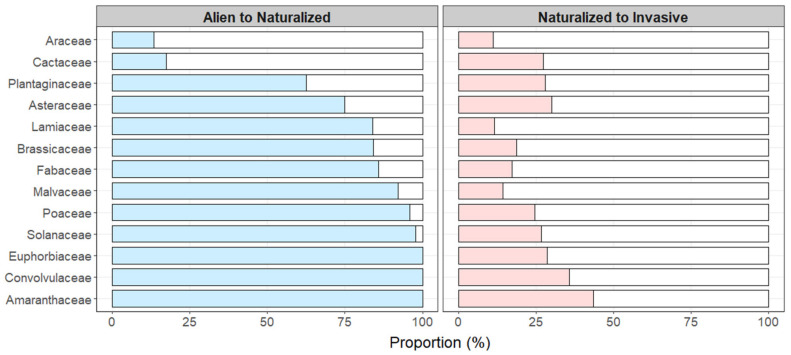
Proportion of species in dominant families that have transitioned from alien to naturalized status, and from naturalized to invasive status.

**Figure 3 plants-10-02241-f003:**
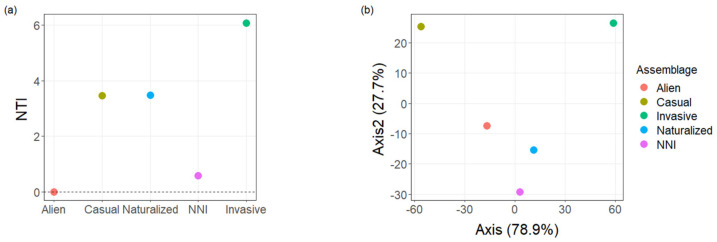
NTI values of different species assemblages: alien, casual aliens, naturalized, NNI (naturalized but not invasive) and invasive species (**a**); and MNTD values between these assemblages (**b**), visualized by PCoA analysis.

## Data Availability

The data are provided in the Appendix A.

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
