# Peer review of "Alien, Naturalized and Invasive Plants in China"

_plants, 2021, doi:10.3390/plants10112241_

Round 1

Reviewer 1 Report

Line 62 The Plant List is no longer active, it is now replaced by World Flora Online (please change reference too).
Line 63 Why do the authors use APG III and not the current APG IV? However add reference.

Lines 62-63 punctuation is not clear

Although the authors considered the classification of alien species according to Pyšek et al., 2004, they did not use it in Figure 2 and throughout the text (i.e. Line 98).

The Authors' answer to the last suggestion of the Reviewer 3 is not clear. What does "there were many mistakes of
authors in the Latin name of species "mean? According to the Reviewer 3 I suggest adding them.

Reviewer 2 Report

The authors improved the manuscript following some of the suggestions of the referees.

I suggest acceptance after the following minor revisions.

The authors used the terms “alien, naturalized and invasive”, in the title and throughout the text. However, the term “alien” includes “casual, naturalized and invasive”. This is confusing. The changed it in some parts of the manuscript, for example indicating “casual” instead of “alien not naturalized”. Ok. But the confusing use of these terms still persists in other parts of the manuscript.

Many species in Table S1 are indicated as “native to China”. The authors indicated in the Results 1686 alien vascular plant species, while 2277 taxa are reported in Table S1 (in fact, 1686 are indicated as “alien we used” and 591 as “native to China”). The authors should explain (in Material and Methods paragraph) why those 591 taxa considered as native are reported in the Table. For example, as they wrote in the answer letter: “Some species were identified as alien species before. But it turned out to be Native species to China. We put them in Table S1 and marked them as “Native to China” to tell the readers that why it’s not alien species.”

Line 63: Add a reference for “families are organized based on APG III”

Line 92: “Are” instead of “were” and add “vascular”, ie: There are 1686 alien vascular plant species

Lines 93-95: Why “were” and not “are/are recognized”?

Lines 94-95: Add “and 488 were casual alien species (alien but not naturalized species, belonging to 252 genera and 70 families)” at the end of the sentence (line 96)

Table S1: Correct “Alein” to “Alien” (92 times in column J)

Table S1: I suggest adding a column (column Q) for “Casual” species (the 488 taxa with 1 0 0 in columns N O P, respectively)

Reviewer 3 Report

The authors reviewed families, lifeforms, origins, introduction pathways and phylogenetic diversity of alien, naturalized and invasive vascular plants in China. The authors used NTI to quantity phylogenetic diversity. Because NTI is a tip-weighted phylogenetic metric, which does not capture deep evolutionary information in the phylogenetic tree, using it alone in this study may not be appropriate. I suggest the authors also use Net Relatedness Index (NRI) in the study.

Author Response

Comments and Suggestions for Authors

The authors reviewed families, lifeforms, origins, introduction pathways and phylogenetic diversity of alien, naturalized and invasive vascular plants in China. The authors used NTI to quantity phylogenetic diversity. Because NTI is a tip-weighted phylogenetic metric, which does not capture deep evolutionary information in the phylogenetic tree, using it alone in this study may not be appropriate. I suggest the authors also use Net Relatedness Index (NRI) in the study.

Response: We appreciate the positive comments, and thank you for the opportunity to revise the manuscript. During the data analysis process, we also calculated NRI (net relatedness index, measures the standardized effect size of MPD between all possible pairs of species in an assemblage). The first two PCoA axes for MNTD explained more than 95% of variance, while MPD was weakly represented by PCoA (the first two PCoA axes explained 54.1% of variance). Thus, we only used NTI (calculated by MNTD) for all the analysis. To make it clear, we add explanations in Lines 93-97.

Round 2

Reviewer 3 Report

none

This manuscript is a resubmission of an earlier submission. The following is a list of the peer review reports and author responses from that submission.

Round 1

Reviewer 1 Report

L 60-64 : some mis-editing due to URL addresses

Phylogenetic assessment should be better described by presenting the phylogenetic tree used for further analyses in an appropriate simplified figure : a phylogenetic tree with the 137 families should be designed and, the numbers of genera per family mentioned.

The legend of the Figure 1 sould be citing the Table S1 for better understanding of the abbreviations and categories used. 

Table S1
Abbrevaitions (table Read Me) sould be corrected by Abbreviations
Referneces (table References) should be corrected by References

As a result of this study, a potential list of the 10 naturalized plant species with a high risk to become invasive in China should be done for a preventive social, economic and, ecological management. 

Reviewer 2 Report

It is not clear the classification used by the authors of alien species in Alien, Naturalized and Invasive. In this way it seems that naturalized and invasive are not alien species. Furthermore, the results chapter is very difficult to interpret.

According to Pyšek et al., 2004 (see the reference below) I suggest to use the following classification for alien plants:

Casual

Naturalized

Invasive

Pyšek, P., Richardson, D. M., Rejmánek, M., Webster, G. L., Williamson, M., & Kirschner, J. (2004). Alien plants in checklists and floras: towards better communication between taxonomists and ecologists. Taxon, 53(1), 131-143.

Line 51 Could the authors give some examples of ecological disasters due to invasions of alien plants?

Line 64 Could the Authors specify if families are organized based on APG IV or other systems?

Lines 95-96 Following the Pyšek classification the remaining 256 species are casual.

Line 97 Could the authors specify the percentage or number of alien species for each of these family? It is not clear whether they refer to the total number of alien species or to the non-naturalized and non-invasive one.

Figure 1 b In my opinion the use of Raunkiaer's life forms would have made the data analysis easier and the graph more readable

Figure 1 c - Please change T-Asia to Trop Asia

Figure 1 c - What do the authors mean by Artificial? Do they refer to the Cultivars?

Figure 1 d - How is hybridization a pathway? Could the authors give an example?

Figure 1 d - Is it possible to discriminate between Naturally, Unintentional and Unknown? I suggest to bring together the latest 4 categories in a generic Others.

Lines 176-181 - In my opinion the Phylogenetic analysis needs further studies and insights, also analyzing the evidence. I suggest deleting it and devoting further work to this topic.

Reviewer 3 Report

The manuscript deals with the alien vascular plants in China. The topic is interesting. The results and discussion are clearly presented. The English is ok. However, I have a major concern, i.e. the reported numbers seems to be to low. I had expected many more alien taxa in a territory such China. For example, the alien taxa in Italy (which is much smaller than China) were recently (Galasso et al. 2018 - Plant Biosystems 152 (3): 556-592) reported as follows: “1597 species, subspecies, and hybrids, distributed in 725 genera and 152 families; 2 taxa are lycophytes, 11 ferns and fern allies, 33 gymnosperms, and 1551 angiosperms.157 taxa are archaeophytes and 1440 neophytes. The alien taxa currently established in Italy are 791 (570 naturalized and 221 invasive), while 705 taxa are casual aliens.”

Maybe the dataset the authors used is incomplete. They collected data only from literature and online. I think that botanists working in field should be involved and the data re-analyzed after having a “complete” dataset of vascular plants alien to China.

The bibliographic citations do not follow the journal rules. For example, in the text do not write “(Prentis et al. 2008; Mollot et al. 2017; Ying et al. 2017)” (line 34) but “[1,2,3]” and in the references list (Lines 211-272) do not follow alphabetic order but follow the numbering of citations in the text.

Minor points:

Line 15: Add “vascular”, i.e. “and invasive vascular plants in China”

Line 58: Add “vascular”, i.e. “invasive vascular plants”

Line 78: “Tsirogiannis and Sandel, 2017” instead of “Tsirogiannis, 2017”

Line 82: Check the sentence. Maybe “randomly/randomized assemblaged.” instead of “randomized assemblages.”

Line 136: “were” instead of “was”, i.e. “Invasive species were”

Line 144: Add “among”, i.e. “they are among the biggest” (In fact, Orchidaceae are bigger as Poaceae and Fabaceae)

Line 177: Probably you mean “and also from alien not naturalized species”

Line 221: “Divisek” instead of “Diviek”

Online supplement (Table S1): Many species are indicated as “Native to China”. Why are those taxa in this Table? Were they considered as alien taxa before? Explain, please.

Online supplement (Readme, column B, line 1): “Abbreviations” instead of “Abbrevaitions”

Online supplement (Table S1): I suggest to follow the following sequence of columns: Phylum – Family – Genus

Online supplement (Table S1, column D): You indicated a nomenclatural reference in the Material and Methods. Nevertheless, I suggest to add the author(s) to each taxon in column D (lines 2-2278)

Reviewer 4 Report

Biological invasions are among the greatest ecological problems today. However, the authors did not sufficiently justify the research undertaken here in the introduction. It is a short report from a database compiled by the authors rather than a research paper. The conducted analyses do not allow for predicting the possibility of invasion, let alone preventing it. The Introduction and Discussion should refer to more important publications on biological invasions and phylogenetic relationships between native and alien species. There are no reasonable research hypotheses here. The research goals also seem to be unrealized. Hence, the Discussion is very weak, and the conclusions are too hasty and little related to the research results. However, I believe that this material has some potential, but it requires a lot of effort on the part of the authors. It won't get better in a few days. Therefore, I suggest "reject and resubmission".